# GePBench: Evaluating Fundamental Geometric Perception for Multimodal Large Language Models

## Abstract

Geometric shapes play important roles in both physical world and human cognition. While multimodal large language models (MLLMs) have made significant advancements in visual understanding, their abilities to recognize geometric shapes and their spatial relationships, which we term *geometric perception*, are not explicitly and systematically explored. To address this gap, we introduce GePBench, a novel benchmark specifically designed to assess the geometric perception capabilities of MLLMs. Our extensive evaluations reveal that even the current state-of-the-art MLLMs exhibit significant deficiencies in geometric perception tasks. Furthermore, we show that models trained with GePBench data demonstrate considerable improvements on a wide range of downstream tasks, highlighting the critical role of geometric perception in enabling advanced multimodal applications. Our code and datasets will be publicly available.

## 1 Introduction

Geometric shapes are foundational elements in both natural and artificial environments (Tommasi et al., 2012). In scientific and engineering disciplines, geometric representations enable precise modeling and problem-solving (Berg et al., 2008); in everyday contexts, they support navigation, design, and visual communication (Manippa & Tommasi, 2023). Crucially, geometric shapes act as a bridge between sensory perception and abstract reasoning, forming a universal framework through which humans interpret and structure their surroundings.

Recent advancements in multimodal large language models (MLLMs) have demonstrated remarkable performance across a broad spectrum of tasks (OpenAI, 2023; Google, 2023; Chen et al., 2024c), including scene understanding (Hudson & Manning, 2019), visual commonsense reasoning (Fu et al., 2024), and expert-level visual question answering (Yue et al., 2024). However, a critical question remains largely underexplored: *how well do MLLMs perceive and recognize geometric shapes and their spatial relationships?* We refer to this capability as *geometric perception*, i.e., the ability to recognize shapes, comprehend spatial configurations, and understand structural relationships. Geometric perception is essential for MLLMs, as it lays the foundation for a wide range of downstream applications. For instance, tasks such as medical image analysis (Chen et al., 2024a; Yan et al., 2024; Khan et al., 2024) and fossil classification (Barucci et al., 2024; Hou et al., 2023) heavily rely on accurate spatial awareness and the ability to discern abstract geometric patterns.

To systematically evaluate the geometric perception capability of MLLMs, we present GePBench in this work. Our dataset is constructed from our specialized data synthesis engine that generates structured textual descriptions, which are then translated into geometric figures. From these figures, multiple-choice questions and answers are systematically created, ensuring a rigorous and diverse evaluation framework. GePBench comprises 80K images and 285K questions, categorized into easy and hard levels, and evaluates 6 key aspects of geometric perception: location, size, existence, counting, reference, and relationships. Figure 1 shows examples for these aspects.

While several prior datasets, including GeoQA (Chen et al., 2021), Geometry3K (Lu et al., 2021), UniGeo (Chen et al., 2022), geomVerse-V0 (Kazemi et al., 2023), GeoMM (Deng et al., 2024), and MAVIS-Instruct (Zhang et al., 2024b), also involve geometric figures, their primary focus lies in mathematical reasoning tasks, including numeric calculations, proof generation, and relationship

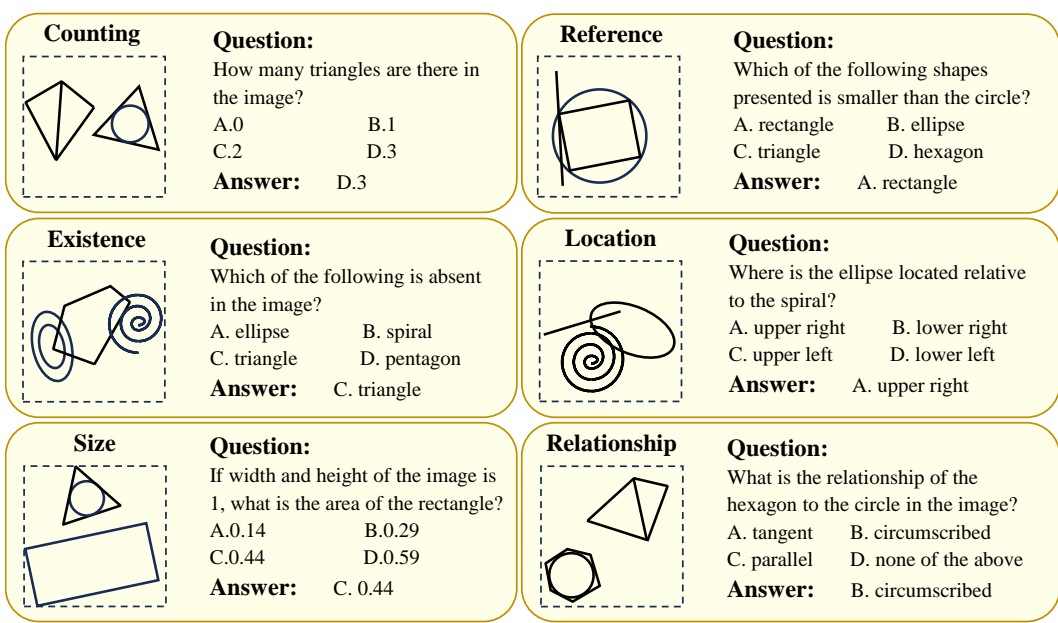

Figure 1: Examples for the different aspects of GePBench.

inference. These higher-order tasks implicitly depend on basic geometric perceptual skills including spatial awareness and shape recognition, which are often insufficiently addressed. In contrast, GePBench explicitly targets geometric perception, providing a more focused evaluation.

We conduct extensive evaluations of GePBench using a diverse array of MLLMs, encompassing both closed-source and open-source models. The results consistently reveal significant limitations in geometric perception. While humans achieve near-perfect accuracy on these tasks with minimal effort, leading models like GPT-4o and Qwen2.5-VL-72B struggle significantly. On tasks such as determining the size of a geometric shape, they achieve accuracies as low as 16.1% and 15.5%, below random guessing. These results underscore the limitations of current MLLMs in basic geometric perception, highlighting the urgent need for advancements in foundational geometric understanding.

Furthermore, we propose LLaVA-GeP, an enhanced model based on the LLaVA architecture trained with data generated by our specialized synthesis engine. LLaVA-GeP demonstrates considerable improvements across various tasks. For example, it achieves an average performance boost of 2.1% on medical image analyses, and 2.4% on chart and document understanding. This underscores the pivotal role that geometric perception plays in enabling broader real-world applications and highlights the transferability of these foundational skills to more complex domains.

Our contributions are summarized as follows:

1) We underscore the significance of geometric perception and introduce GePBench, a novel benchmark focusing on this fundamental perception ability of MLLMs.
2) We conduct extensive evaluations with 27 state-of-the-art models, identifying key technical challenges in geometric perception and providing insights into potential improvements.
3) We propose LLaVA-GeP, a model with upgraded visual capabilities, showing the potential to improve performance in real-world applications through enhanced geometric perception.

## 2 RELATED WORK

### 2.1 MULTIMODAL LARGE LANGUAGE MODELS

In recent years, MLLMs have gained considerable attention for their ability to perform cross-modal understanding across a wide range of real-world tasks (Liu et al., 2023; Zhu et al., 2024; Yao et al.,

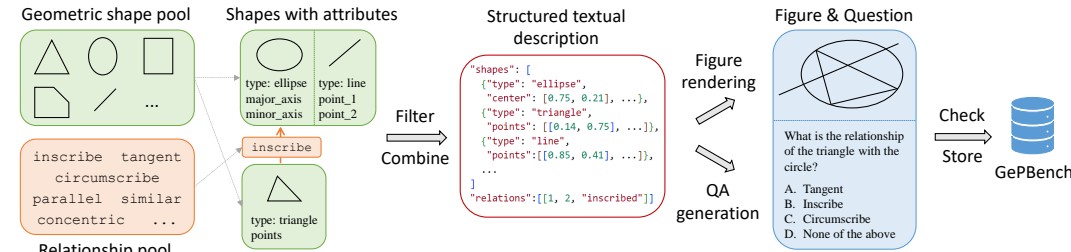

Figure 2: An overview of the data engine of GePBench.

2024; Chen et al., 2023b; Lu et al., 2024a; Tong et al., 2024a). These models typically consist of three main components: a visual encoder responsible for encoding the input image, a language model that enables textual understanding and reasoning, and a mapping module that translates visual features into textual representations. While they have shown remarkable success in various downstream applications, their performance on perceiving and recognizing geometric shapes and spatial relationships remains underexplored. This work aims to investigate whether MLLMs can perform well in geometric perception tasks.

## 2.2 MULTIMODAL BENCHMARKS

The evaluation of MLLMs has been supported by a wide array of multimodal benchmarks, which primarily focus on assessing high-level vision capabilities. Early benchmarks like VQAv2 (Goyal et al., 2019), GQA (Hudson & Manning, 2019), and ScienceQA (Lu et al., 2022) were designed to evaluate specific tasks, including object identification, optical character recognition, and scientific diagram comprehension. Subsequent datasets extended these evaluations to encompass more complex reasoning tasks, such as commonsense reasoning (Luo et al., 2024; Chen et al., 2024b) and professional knowledge assessment (Yue et al., 2024; Lu et al., 2024b). Some benchmarks like MMBench (Liu et al., 2024b) and SeedBench (Li et al., 2023a), introduced hierarchical evaluation frameworks that assess both understanding and reasoning abilities across multiple levels of complexity.

A separate line of research has explored the mathematical reasoning capabilities of MLLMs, with some of them focusing on geometry-related tasks. Early works (Chen et al., 2021; Lu et al., 2021; Chen et al., 2022; Lu et al., 2024b) adapted images from mathematical textbooks or exams, creating questions centered on reasoning tasks like calculations, proof generation, and relationship inference. More recent approaches have automated the construction of these tasks (Deng et al., 2024; Zhang et al., 2024b). Nevertheless, these benchmarks primarily target long-chain mathematical reasoning and often overlook basic geometric perception, leaving a critical gap in our understanding of MLLMs' foundational visual capabilities.

## 3 GEPBENCH

GePBench is a novel benchmark designed to evaluate the geometric perception capabilities of MLLMs. It leverages a data synthesis engine to generate structured textual descriptions of geometric figures, from which corresponding images and multiple-choice questions are constructed. Figure 2 shows an overview of the GePBench data engine.

### 3.1 STRUCTURED DESCRIPTION GENERATION

The foundation of GePBench lies in generating structured textual descriptions, which serve as the basis for both figure creation and question-answer generation. This process ensures consistency and precision in data construction.

To build a comprehensive dataset for evaluating geometric perception, we begin by curating a pool of 15 commonly encountered basic geometric shapes (e.g., lines, polygons, ellipses). From this pool, shapes are randomly sampled and assigned geometric attributes, including size, position, and orientation, subject to type-specific constraints designed to ensure plausibility and avoid ambiguity.

For instance, ellipses are required to have a major-to-minor axis ratio of at least 1.2 to prevent near-circular degeneracy. Overlaps between distinct shape types are restricted on a case-by-case basis to avoid the emergence of unintended composite forms that could complicate annotation.

To simulate real-world scenarios involving spatial interaction among multiple shapes, we define a relationship pool comprising 11 common geometric relations (e.g., tangency, parallelism, inscription). These relationships are probabilistically sampled and geometrically enforced during the incremental construction of each figure. The outcome of this process is a structured textual description that includes shape types, attributes, and spatial relationships between shapes. Further implementation details and constraint specifications are provided in Appendix B.

## 3.2 FIGURE RENDERING

The structured descriptions are then rendered into visual figures using the Matplotlib package (Hunter, 2007) in Python. To align with real-world conditions where noise might exist in collected images, we incorporate some visual noise into the figures with a probability of 0.5. These include Gaussian noise for the background, random disturbance and salt-and-pepper noise on shape outlines, and Perlin noise (Perlin, 1985) for closed shapes. Examples can be found in Appendix G.2. The added noise improves the fidelity of the benchmark by simulating real-world scenarios, leaving challenges for MLLMs under visually degraded conditions.

## 3.3 QUESTION-ANSWER GENERATION

Once figures and textual descriptions are generated, template-based pipelines construct multiple-choice questions across six key aspects, including existence, counting, location, size, reference and relationship. The selection of six evaluation aspects is grounded in both geometric attributes and object attributes, ensuring comprehensive coverage of geometric perception. Size and location are intrinsic properties of geometric objects[1], while relationships capture interactions between multiple shapes. Existence, counting and reference represent key evaluative dimensions derived from object attributes, which are widely recognized in other established visual benchmarks. For example, POPE (Li et al., 2023c) includes object existence to assess hallucinations, and SeedBench (Li et al., 2023a) incorporates instance counting and visual reference to evaluate models' visual recognition capability. By encompassing these six aspects, GePBench assesses core geometric perceptual capabilities, including shape recognition, relationship understanding, and spatial awareness, providing a comprehensive evaluation framework.

Specifically, for any given figure, a question-answer pair is constructed by randomly selecting a shape and formulating a question based on its geometric attributes. The ground truth answer is derived from the structured textual description of the figure, ensuring accuracy and consistency. To create multiple-choice questions, we generate plausible distractors, i.e., incorrect answers designed to challenge the model's geometric perception abilities, and combine them with the correct answer into a set of four candidate choices. Finally, they are categorized into easy and hard levels according to whether the number of shapes is greater than four and whether noise is added in the figure. Details on different aspects and their associated generation strategies can be found in Appendix B.3.

## 3.4 STATISTICS AND ANALYSIS

According to the statistics, GePBench contains a total of 80K geometric figures and 285K multiple choice questions. The dataset sizes for 6 aspects are outlined in Figure 3 (b). GePBench features a diversity of geometric shapes and question types. Detailed statistics are presented in Figure 3 (a) (c), including the number of shapes per figure and question length, highlighting its balanced coverage across geometric types and question categories.

To validate the quality of the dataset, we conducted human evaluation with 20 participants from different professions and age groups. Each participant was asked to solve a subset containing 200 questions for both the easy and hard splits of GePBench. The results, summarized in Table 1, reveal an average human accuracy of 99.3%, demonstrating that the tasks are intuitive and solvable for humans. Details are available in Appendix C.3.

---

[1]https://en.wikipedia.org/wiki/Shape

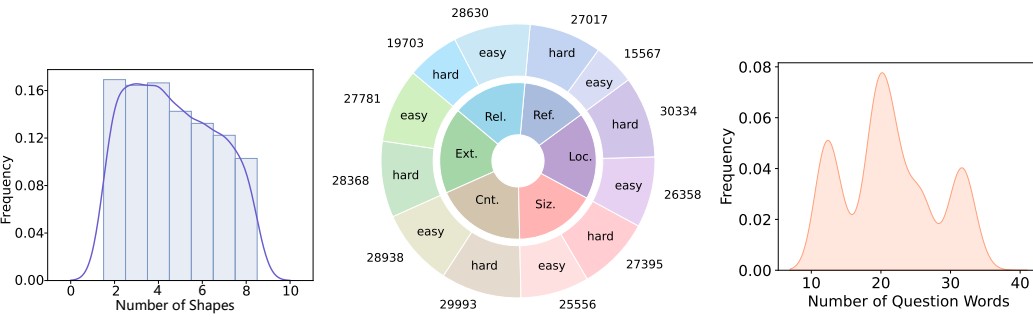

(a) Number of shapes per figure.  (b) Sizes of different aspects.  (c) Number of words per question.

Figure 3: Key data distributions of GePBench.

Overall, GePBench provides a rigorous and diverse evaluation benchmark, challenging MLLMs with foundational geometric perception.

## 4 EXPERIMENTS

### 4.1 EXPERIMENTAL SETUP

**Evaluated models.** We perform a comprehensive evaluation on 27 multimodal LLMs, which are divided into 3 groups: closed-source models, including GPT-4o (OpenAI, 2023) and Gemini-2.5-Pro (Google, 2023); open-source general-purpose models, including BLIP2 (Li et al., 2023b), InstructBLIP (Dai et al., 2023), MiniG-PTv2 (Chen et al., 2023a), LLaVA-1.5 (Liu et al., 2024a), LLaVA-OneVision (Li et al., 2024), mPLUG-Owl3 (Ye et al., 2024), InternVL series (Chen et al., 2024c) (Zhu et al., 2025), MiniCPM-V-2.6 (Yao et al., 2024), GLM-4V (GLM et al., 2024), Mantis-Idefics2 (Jiang et al., 2024a), Qwen-VL series (Wang et al., 2024b) (Bai et al., 2025), LLaMA-3.2-Vision (Meta, 2024); and specialized geometric and reasoning models, including G-LLaVA (Gao et al., 2025), Math-LLaVA (Shi et al., 2024), Math-PUMA (Zhuang et al., 2025). Appendix C.1 shows the details of these models.

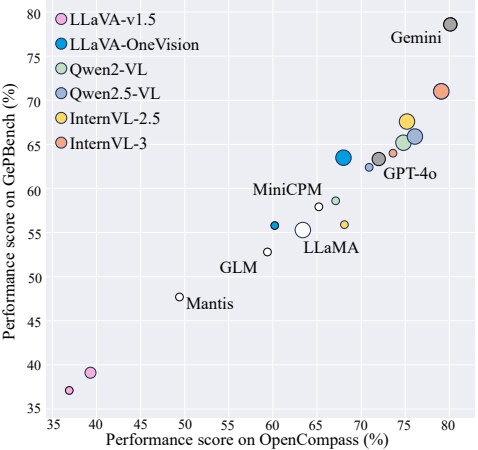

Figure 4: Performance comparison of representative models on GePBench and OpenCompass. Larger dots indicate larger model sizes in the same series. The scores of GePBench and OpenCompass align considerably well.

**Evaluation setup.** All evaluations are conducted exclusively in a zero-shot manner. For a fair comparison, the temperature is set to 0 and the image resolution is set to 640x640 for all the models. Accuracy is adopted as the metric for each aspect. Detailed evaluation schemes and corresponding prompts are provided in Appendix C.2.

### 4.2 MAIN RESULT

The main experimental results are presented in Table 1. To compare the performance of the MLLMs on fundamental geometric perception and high-level multimodal semantic tasks, we also include evaluation results reported by OpenCompass (Contributors, 2023), which are visualized in Figure 4. Our analysis reveals the following key observations:

**Most models, especially open-source ones, face considerable challenges on GePBench.** The results of Table 1 indicate that most MLLMs encounter significant challenges when evaluated on geometric perception tasks. For advanced closed-source models, Gemini-2.5-pro demonstrates

Table 1: Performance comparison of different MLLMs on GePBench (%). Ext, Cnt, Siz, Loc, Ref, Rel represent existence, counting, size, location, reference, relationship, and Avg is the average value of all the 12 following scores. Best scores are in bold.

| Model Class | Size | Avg. | Easy | | | | | | Hard | | | | | |
|---|---|---|---|---|---|---|---|---|---|---|---|---|---|---|
| | | | Ext. | Cnt. | Siz. | Loc. | Ref. | Rel. | Ext. | Cnt. | Siz. | Loc. | Ref. | Rel. |
| Random guessing | - | 25.0 | 25.0 | 25.0 | 25.0 | 25.0 | 25.0 | 25.0 | 25.0 | 25.0 | 25.0 | 25.0 | 25.0 | 25.0 |
| Human | - | 99.3 | 99.4 | 99.8 | 99.2 | 98.4 | 99.9 | 99.6 | 99.3 | 99.7 | 98.9 | 98.5 | 99.8 | 98.8 |
| GPT-4o | - | 63.2 | 77.4 | 70.5 | 16.1 | 61.5 | 88.3 | 84.8 | 73.3 | 63.5 | 18.1 | 65.2 | 74.2 | 65.0 |
| Gemini-2.5-pro | - | **78.6** | 79.0 | 77.8 | **73.1** | **80.3** | 86.1 | **88.2** | 70.8 | 75.0 | 78.5 | 76.8 | 72.0 | **86.0** |
| BLIP2 | 3B | 33.9 | 35.4 | 18.4 | 26.9 | 27.2 | 48.2 | 52.3 | 41.0 | 15.9 | 34.5 | 25.0 | 37.9 | 44.1 |
| InstructBLIP | 3B | 33.5 | 36.9 | 23.2 | 21.2 | 27.7 | 54.7 | 49.4 | 42.1 | 23.6 | 17.5 | 25.9 | 42.9 | 37.1 |
| MiniGPTv2 | 7B | 28.9 | 24.6 | 35.3 | 21.2 | 31.5 | 27.7 | 38.8 | 28.2 | 32.2 | 26.0 | 26.3 | 25.8 | 28.7 |
| LLaVA-1.5 | 7B | 37.1 | 33.8 | 44.4 | 20.2 | 41.3 | 36.5 | 67.5 | 32.8 | 25.5 | 22.0 | 32.1 | 38.5 | 50.3 |
| LLaVA-1.5 | 13B | 39.1 | 41.5 | 55.6 | 5.7 | 40.4 | 36.5 | 69.6 | 42.1 | 30.3 | 15.3 | 39.3 | 44.0 | 48.3 |
| mPLUG-Owl3 | 7B | 47.5 | 53.3 | 66.2 | 15.5 | 34.7 | 62.0 | 69.6 | 57.4 | 51.4 | 22.6 | 31.2 | 54.9 | 51.0 |
| MiniCPM-V-2.6 | 8B | 57.9 | 64.1 | 73.9 | 31.1 | 45.5 | 75.2 | 74.3 | 65.1 | 56.2 | 30.5 | 55.4 | 68.1 | 55.2 |
| GLM-4V | 9B | 52.8 | 54.9 | 72.9 | 20.7 | 44.6 | 83.2 | 70.9 | 53.3 | 51.4 | 13.6 | 53.1 | 69.8 | 45.5 |
| Mantis-Idefics2 | 8B | 47.7 | 54.4 | 64.3 | 17.6 | 43.7 | 67.2 | 62.0 | 55.9 | 47.1 | 14.7 | 41.5 | 58.2 | 46.2 |
| LLaMA-3.2-Vision | 90B | 55.4 | 62.6 | 72.0 | 13.0 | 49.3 | 69.3 | 82.7 | 60.0 | 54.8 | 16.9 | 51.8 | 67.6 | 65.0 |
| LLaVA-OneVision | 7B | 55.8 | 62.6 | 74.9 | 26.9 | 53.1 | 74.5 | 72.2 | 60.0 | 56.7 | 23.7 | 54.0 | 57.7 | 53.1 |
| LLaVA-OneVision | 72B | 63.6 | 73.3 | 78.7 | 24.4 | 53.1 | **93.4** | 84.8 | 68.7 | 58.2 | 24.3 | 66.1 | 75.8 | 62.9 |
| Qwen2-VL | 7B | 58.6 | 67.2 | 80.2 | 20.2 | 60.6 | 78.8 | 76.8 | 65.1 | 61.1 | 11.9 | 53.6 | 73.6 | 53.8 |
| Qwen2-VL | 72B | 65.3 | 70.3 | 75.4 | 22.8 | 77.0 | 84.7 | 81.0 | 69.2 | 65.9 | 17.5 | **76.8** | 76.9 | 65.7 |
| Qwen2.5-VL | 7B | 62.4 | 71.8 | 75.4 | 25.9 | 68.5 | 81.0 | 78.9 | 66.7 | 61.1 | 21.5 | 71.0 | 70.9 | 56.6 |
| Qwen2.5-VL | 72B | 66.0 | 75.4 | 76.8 | 15.5 | 70.0 | 86.9 | 82.3 | 71.3 | 66.3 | 30.5 | 69.6 | 79.1 | 68.5 |
| InternVL2.5 | 8B | 55.9 | 61.5 | 72.9 | 17.1 | 61.5 | 55.5 | 80.2 | 62.6 | 58.7 | 14.1 | 59.8 | 58.8 | 67.8 |
| InternVL2.5 | 78B | 67.7 | 76.9 | **81.2** | 21.8 | 70.4 | 92.0 | 84.4 | 71.3 | 66.3 | 30.5 | 69.6 | 79.1 | 68.5 |
| InternVL3 | 8B | 64.0 | **80.0** | 73.4 | 22.8 | 67.1 | 81.8 | 81.0 | **73.8** | 56.2 | 31.6 | 71.0 | 75.3 | 53.8 |
| InternVL3 | 78B | 71.0 | 75.4 | 76.3 | 50.3 | 65.7 | 86.1 | 84.8 | 73.3 | 66.3 | 48.0 | 75.9 | **81.3** | 68.5 |
| G-LLaVA | 7B | 26.5 | 23.6 | 22.2 | 20.2 | 26.8 | 29.2 | 42.6 | 22.1 | 20.7 | 32.2 | 27.2 | 24.2 | 27.3 |
| G-LLaVA | 13B | 29.2 | 23.6 | 31.4 | 26.9 | 25.4 | 36.5 | 39.2 | 29.2 | 26.9 | 26.6 | 25.9 | 34.1 | 25.2 |
| Math-LLaVA | 13B | 38.2 | 46.2 | 40.6 | 12.4 | 33.3 | 56.2 | 59.5 | 46.2 | 21.2 | 19.2 | 34.4 | 47.3 | 42.0 |
| Math-PUMA | 7B | 44.3 | 42.1 | 56.0 | 31.6 | 22.1 | 59.9 | 74.3 | 45.1 | 33.7 | 27.7 | 28.6 | 60.4 | 49.7 |
| QVQ-preview | 72B | 57.0 | 67.2 | 67.6 | 22.3 | 49.8 | 73.0 | 73.8 | 67.2 | 65.9 | 31.1 | 53.6 | 59.9 | 53.1 |

relatively satisfactory performance across various aspects, whereas GPT-4o lags behind, especially on size estimation aspect of the easy split with an accuracy of only 16.1%. In comparison, open-source models perform even worse overall. Notable exceptions include LLaVA-OneVision, InternVL series and Qwen-VL series, whose performance is comparable to closed-source models. However, the majority of open-source models fall below 60%. This disparity underscores the limitations of open-source models in geometric perception and the substantial room for improvement. Our findings underscore the challenges of geometric perception and highlight its value in testing the limitations of state-of-the-art systems.

**Size and Location are generally more challenging than other aspects for current MLLMs.** Among geometric perception tasks, size and location prove the most difficult for both model types, reflecting deficiencies in spatial awareness. As shown in Table 1, even the top-performing open source model InternVL3-78B achieves only 50.3% accuracy on size-related questions and 65.7% on location-related questions in the easy split. Many other models fare worse, often falling below random guessing. This struggle likely stems from the design of modern visual encoders, which prioritize robustness in real-world image understanding by enforcing invariance to transformations like cropping and rotation (Anwar et al., 2015). While beneficial for general image recognition, these properties hinder precise spatial perception (Tu et al., 2024). Addressing this issue requires rethinking training paradigms to balance invariance with sensitivity to spatial details.

**Specialized geometric and reasoning models do not significantly outperform their base models on GePBench.** Models specifically designed for geometric or mathematical reasoning, such as G-LLaVA and Math-PUMA, all fail to surpass their general-purpose counterparts (LLaVA-1.5 and Qwen2-VL, respectively). This outcome likely stems from the fact that these models are primarily trained on datasets tailored for mathematical problem-solving, neglecting the foundational challenges of visual perception and spatial awareness. Consequently, they struggle to generalize to the perceptual demands of GePBench. Similarly, the visual reasoning model QVQ-preview-72B fails to outperform

Table 2: Performance comparison of LLaVA-1.5-7B with different visual encoders (%).

| Encoder class | Resolution | Avg. | Easy | | | | | | Hard | | | | | |
|---|---|---|---|---|---|---|---|---|---|---|---|---|---|---|
| | | | Ext. | Cnt. | Siz. | Loc. | Ref. | Rel. | Ext. | Cnt. | Siz. | Loc. | Ref. | Rel. |
| CLIP | $224^2$ | **37.4** | 33.3 | 44.0 | 19.2 | **47.4** | 28.5 | 65.8 | 31.8 | **26.0** | 22.6 | **44.2** | 37.4 | 49.0 |
| CLIP | $336^2$ | 37.1 | 33.8 | 44.4 | 20.2 | 41.3 | 36.5 | **67.5** | 32.8 | 25.5 | 22.0 | 32.1 | **38.5** | **50.3** |
| OpenCLIP | $224^2$ | 37.3 | 31.8 | 38.2 | 21.2 | 46.5 | **37.2** | 67.1 | 32.8 | 22.1 | 22.0 | 42.0 | 37.9 | 49.0 |
| DINOv2 | $224^2$ | 31.1 | 30.8 | 33.3 | 19.7 | 30.0 | 21.9 | 64.6 | 32.3 | 16.3 | 19.2 | 29.0 | 30.8 | 45.5 |
| SigLIP | $224^2$ | 37.0 | **34.9** | **46.9** | 21.2 | 42.3 | 28.5 | 66.7 | **37.4** | 20.7 | **25.4** | 37.5 | 34.6 | 47.6 |
| CLIP + DINOv2 | $224^2 + 224^2$ | 33.4 | 29.7 | 38.6 | 18.1 | 35.2 | 24.8 | 63.3 | 32.8 | 18.8 | 20.9 | 34.8 | 35.7 | 48.3 |
| CLIP + DINOv2 | $336^2 + 224^2$ | 31.9 | 30.3 | 27.5 | **22.3** | 27.2 | 27.0 | 66.2 | 33.3 | 13.0 | 24.9 | 28.1 | 34.1 | 48.3 |

its base model, Qwen2-VL-72B. This observation aligns with the findings in QVQ's technical report[2], which states: "QVQ doesn't show significant improvement over Qwen2-VL-72B in basic recognition tasks." Since GePBench primarily assesses fundamental visual perception capabilities, advanced reasoning mechanisms offer little advantage in this context.

## 5 DISCUSSION

In this section, we present further systematic analyses of geometric perception in MLLMs. We investigate the influence of visual encoders on model performance (Section 5.1), the benefits of geometric perception for downstream tasks (Section 5.2), and the performance gap between models and humans on various benchmarks (Section 5.3). We also conduct a qualitative error analysis to gain deeper insights into the common failure modes of MLLMs (Section 5.4). Additionally, we provide more discussions in the Appendix, including impact of the number of shapes on model performance in Appendix G.1, and detailed noise analyses comparing figures with different level of noise in Appendix G.2.

### 5.1 IMPACT OF VISUAL ENCODERS

As discussed in Section 2.1, visual encoders are central to the visual perception capabilities of MLLMs. To assess their influence, we experiment with the LLaVA-1.5-7B model using various visual encoders, including CLIP-ViT-L, CLIP-ViT-L-336 (Radford et al., 2021), OpenCLIP-ViT-L (Cherti et al., 2023), DINOv2 (Oquab et al., 2024) and SigLIP (Zhai et al., 2023). Additionally, following the empirical best practice (Tong et al., 2024b; Yang et al., 2024), we evaluate mixed configurations where outputs from both the CLIP-ViT-L and DINOv2 encoders are fed into the LLM backbone. The checkpoints for these models are sourced from Yang et al. (2024) and evaluated on GePBench. The results, summarized in Table 2, provide the following insights:

1) A comparison between CLIP-ViT-L and CLIP-ViT-L-336 suggests that higher resolution enhances fine-grained recognition but may compromise spatial accuracy, as it improves performance in most aspects of geometric perception except for location.
2) Different encoders exhibit distinct strengths, excelling in different aspects of geometric perception as indicated by their varying performance across tasks.
3) Contrary to tasks that require high-level vision semantics capabilities (Tong et al., 2024a; Yang et al., 2024), Mixed encoders underperform in fundamental geometric perception. This may be attributed to the additional complexity of positional embeddings introduced by a larger number of image tokens from multiple encoders, resulting in degraded spatial awareness.

For a more detailed discussion, please refer to Appendix D.

### 5.2 BENEFITS FOR DOWNSTREAM TASKS

To explore the broader impact of geometric perception on downstream multimodal tasks, we utilize the data constructed by our data engine to train LLaVA-GeP-7B, a multimodal large language model based on the LLaVA-1.5-7B architecture.

Specifically, following the two-stage training procedure in LLaVA, we construct additional training data using the data engine of GePBench. We generate captions for the figures as pretraining data

---

[2]https://huggingface.co/Qwen/QVQ-72B-Preview

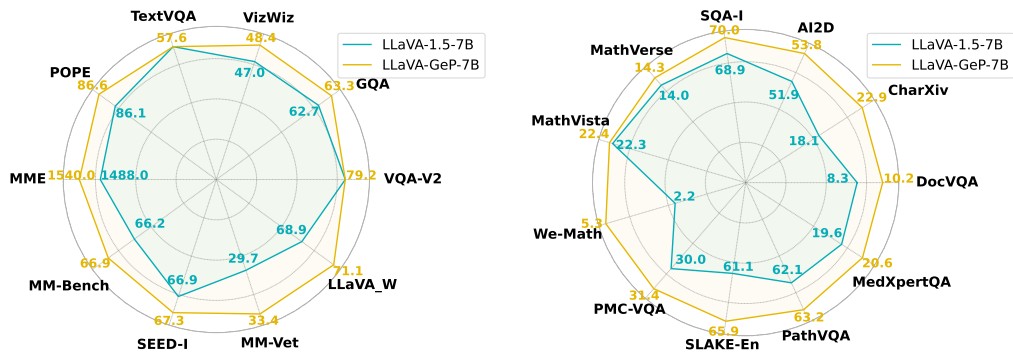

Figure 5: Performance comparison of of LLaVA-1.5-7B and LLaVA-GeP-7B across general datasets (left) and specialized benchmarks (right) on downstream tasks.

in the first stage with the help of LLMs. The multiple-choice questions are directly utilized as instruction-tuning data in the second stage. Finally, 300K and 240K samples are obtained and mixed with the original LLaVA training data for pretraining and instruction tuning respectively.

The trained model, LLaVA-GeP-7B, is evaluated on a wide range of tasks. These tasks encompass both general benchmarks consistent with those used in the original LLaVA evaluation, and specialized benchmarks focused on domains such as mathematics, medical image analyses, and scientific chart and document understanding, which are inherently closer to geometric perception. More information on training, implementation and evaluation are provided in Appendix E.

The results, visualized in Figure 5, show consistent improvements across the evaluated tasks. Notably, general tasks demanding spatial awareness, abstract visual understanding and scientific diagram comprehension, such as MME-Perception and MM-Vet, exhibit considerable gains. A closer analysis of SeedBench and MMBench results reveals that most improvements are concentrated in categories of instance interaction, counting, and spatial localization. Moreover, LLaVA-GeP-7B also achieves better performance on domain-specific tasks, especially in medical imaging and scientific chart understanding. These results suggest that training on geometric perception data enhances the model's ability to discriminate relationships and understand spatial configurations, which benefits effectively real-world scenarios where high-level skills are required.

## 5.3 GAPS BETWEEN MLLM AND HUMAN

To assess how well state-of-the-art MLLMs measure up to human performance, we compare their capabilities across diverse benchmarks, namely A-Bench (Zhang et al., 2024d), BLINK (Fu et al., 2024), CODIS (Luo et al., 2024), HR-Bench (Wang et al., 2024c), II-Bench (Liu et al., 2025), $M^3CoT$ (Chen et al., 2024b), MARVEL (Jiang et al., 2024b), Math-Verse (Zhang et al., 2024a), MathVista (Lu et al., 2024b), MMMU (Yue et al., 2024), MuirBench (Wang et al., 2024a), Q-Bench (Wu et al., 2024), UNIAA (Zhou et al., 2024), VCR (Zhang et al., 2024c), and WinoGround (Thrush et al., 2022). These benchmarks span visual understanding and inference, visual reasoning, detailed understanding of real-world images, mathematics, and other integrated tasks. For consistency, we use performance data from GPT-4o and the human average. When GPT-4o data is unavailable, we substitute with GPT-4V. Detailed data collection methods are outlined in Appendix H.

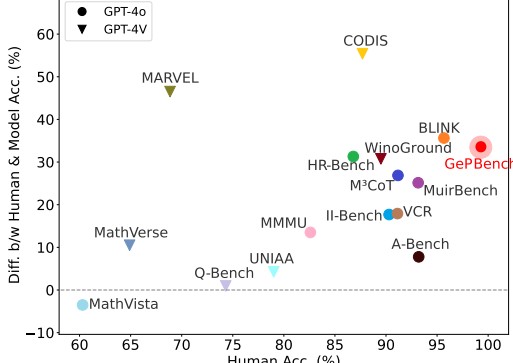

Figure 6: Human performance versus the performance gap between models and humans. Large y-values indicate MLLMs underperform significantly compared to humans.

Figure 6 visualizes the results. The x-axis represents average human scores on various benchmarks, with higher values indicating easier tasks for humans. The y-axis denotes the performance gap between MLLMs and humans, where larger values represent greater MLLM underperformance.

The analysis reveals that geometric perception emerges as a particularly notable task: despite being simplest for humans, it presents a considerable challenge for MLLMs. Humans achieve near-perfect accuracy on the straightforward multiple-choice questions, yet the powerful GPT-4o lag by more than 36%. This pronounced gap underscores the critical need to enhance MLLMs' geometric perception capabilities to bridge this persistent divide.

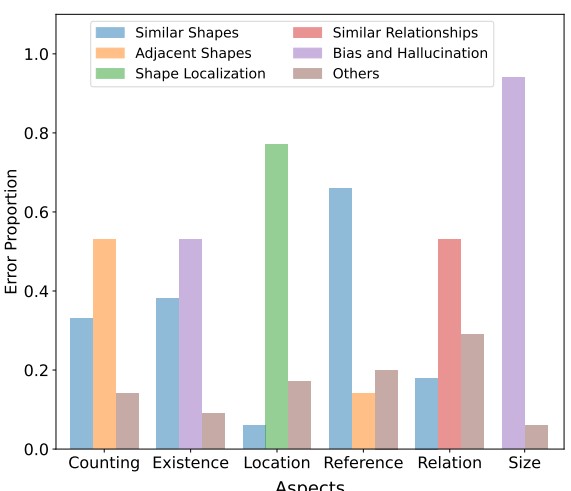

Figure 7: Distribution of error types across different aspects for GPT-4o.

### 5.4 ERROR ANALYSES

To gain deeper insights into the common failure modes of MLLMs in geometric perception tasks, we conduct an error analysis by manually inspecting 200 randomly sampled instances where GPT-4o produces incorrect outputs across different aspects. Our analysis reveals that the majority of errors can be categorized into five primary classes, and their distributions across different aspects are visualized in Figure 7.

1) **Failure to discriminate visually similar shapes.** The model often confuses visually similar shapes (e.g., squares vs. rectangles, spirals vs. circles).
2) **Misinterpretation of adjacent shapes.** Closely positioned or overlapping shapes are frequently miscounted or misidentified, reflecting weak sensitivity to spatial relationships.
3) **Inaccurate localization of geometric shapes.** The model struggles to correctly localize shapes within an image, indicating deficiencies in spatial perception.
4) **Confusion between similar geometric relationships.** Geometric relations such as inscription, circumscription, and tangency are often conflated.
5) **Bias and hallucination.** The model exhibits hallucinations in existence queries and systematic bias in size estimation, often overestimating geometric scales.

In summary, the results further reveal fundamental limitations in current MLLMs' ability to perceive geometric shapes. GPT-4o struggles not only with distinguishing visually similar shapes and relationships but also with accurately interpreting spatial configurations and positional information. Moreover, the prevalence of hallucinations and systematic biases suggests that these models may rely on overgeneralized priors rather than precise visual understanding. Together, these findings highlight critical challenges in visual perception and spatial awareness, which must be addressed to improve the reliability of MLLMs in vision-language tasks. Illustrative examples and detailed description of each error class can be found in Appendix F. There we also provide further analyses, including in-depth investigation into the most prevailing error types Similar Shapes and Bias & Hallucination, and the trace of error sources through module-specific finetuning and probing experiments.

## 6 CONCLUSION

In this paper, we introduce GePBench, a large-scale benchmark dataset specifically designed to evaluate geometric perception for MLLMs. Extensive experiments highlight substantial room for improvement, as even state-of-the-art models fail to achieve satisfactory results. Our analysis of visual encoders provides insights into structural design. Additionally, we demonstrate that enhancing geometric perception contributes to improved performance on a variety of multimodal tasks, underscoring its foundational importance.

REPRODUCIBILITY STATEMENT

To better support reproducibility, we detail all the details to reproduce our results in Appendix C and E, including the evaluation parameters, training details, environment and framework versions.

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

## A  LLM USAGE

In the course of preparing this manuscript and supporting materials, we leveraged large language models (LLMs) as auxiliary tools to enhance the efficiency and quality of non-core research tasks. Specifically, LLMs were employed in two primary capacities:

1. **Language polishing**: We used LLMs to assist in proofreading, grammatical correction, and stylistic refinement of the manuscript's prose.
2. **Boilerplate and utility code generation**: For ancillary implementation tasks, such as file I/O wrappers, format converters, or logging utilities, we used LLMs to accelerate prototyping.

## B  DETAILS ON THE CONSTRUCTION OF GEPBENCH

### B.1  ALGORITHM OVERVIEW

Algorithm 1 describes the entire generation process of structured textual description. This process leverages two predefined asset pools:

1) **The shapes pool** $\mathcal{S}$, includes 15 distinct geometric types: lines, triangles, quadrilaterals, pentagons, hexagons, rectangles, squares, regular pentagons, regular hexagons, pentagram, hexagram, ellipses, circles, sectors and spirals.
2) **The relationships pool** $\mathcal{R}$, consists of 11 fundamental geometric relationships: tangency, parallelism, inscription, circumscription, similarity, concentricity, symmetry, axiality, diagonality, perpendicularity, and adjacency.

The generation process consists of two phases. In the first phase, a sparse set of foundational shapes is generated. The second phase expands upon this set by introducing new shapes that have explicit relationships to existing ones. Before adding a new shape $s$ to the current shape set $S$, the algorithm verifies whether its placement complies with a predefined set of rules. The shape is added to S only if all rules are satisfied. This constraint is crucial for preventing ambiguity and unintended composite forms. The full ruleset is detailed in Appendix B.2.

### B.2  RULESET FOR SHAPES AND ATTRIBUTES

To ensure reasonable attributes when generating structured textual description, we propose a set of rules as guidelines.

---

**Algorithm 1** Structured Description Generation

---

**Require:** Predefined shapes pool $\mathcal{S}$ and relationships pool $\mathcal{R}$, maximum number of shapes per sample $m$, valid placement ruleset $\mathcal{V}$
**Ensure:** A structured description sample $D$
  Initialize the set of shapes $S = \{\}$, the set of relationships $R = \{\}$
  **while** $|S| < \lfloor m/2 \rfloor$ **do**
    $s \leftarrow sample\_new\_shape(\mathcal{S})$
    $randomize\_attributes(s)$
    **if** $is\_placement\_valid(s, S, \mathcal{V})$ **then**
      $S \leftarrow S \cup \{s\}$
  **for** $s$ in $S$ **do**
    $r = sample\_relationship(\mathcal{R}, s)$
    $s^{'} = generate\_shape(r, s)$
    $assign\_attributes(s^{'}, r, s)$
    **if** $is\_placement\_valid(s, S, \mathcal{V})$ **then**
      $S \leftarrow S \cup \{s^{'}\}$
      $R \leftarrow R \cup \{(s, s^{'}, r)\}$
      **if** $|S| \geq m$ **then**
        **break**
  $D \leftarrow (S, R)$
  **return** $D$

---

**Rule 1: Spatial Reasonableness.** Shapes must be of moderate size and fully contained within the canvas.
**Rule 2: Shape Fidelity.** Attributes are constrained to preserve the intended shape category and prevent perceptual ambiguity.
**Rule 3: Topological Simplicity.** Shape intersections are restricted to avoid the formation of unintended shapes.

According to the above rules, we design validation steps when generating specific shapes and employ rejection sampling. The parentheses before each step indicate the corresponding guideline rules.

1) **Polygons.**
   - (Rule 1) The area must exceed 0.02, and each edge length must be greater than 0.05.
   - (Rule 2) All interior angles are constrained to $[0.15\pi, 0.85\pi]$. For rectangles, the height and width must differ by at least 20%.
   - (Rule 3) It should intersect with another polygon, star or line at no more than one point.
2) **Lines.**
   - (Rule 1) Line lengths are constrained to the interval $[0.2, 0.8]$.
   - (Rule 3) A line may intersect a polygon, a star or all other lines at no more than one point.
3) **Stars.**
   - (Rule 1) Outer radius are constrained to the interval $[0.15, 0.4]$.
   - (Rule 2) Inner radius are constrained to 40%-70% of the outer radius.
   - (Rule 3) It should intersect with another polygon, star or line at no more than one point.
4) **Ellipses, Circles, Sectors and Spirals.**
   - (Rule 1) The area must exceed 0.02.
   - (Rule 2) For ellipses, the major and minor axes must differ by at least 20%. For sectors, the angular span are constrained to the interval $\left[\frac{\pi}{12}, \frac{5\pi}{12}\right]$.

By integrating these steps, we systematically eliminate geometrically implausible or semantically ambiguous configurations.

### B.3 DETAILS ON THE DIFFERENT ASPECTS OF GEPBENCH

In this section, we delve into the specifics of the six key aspects of GePBench, highlighting the design principles and generation strategies.

**Existence.**   Questions in this category assess a model's ability to accurately identify whether specific shapes are present in a figure, while avoiding hallucinations or false positives. Rather than relying on simple binary queries (e.g., "Is a circle present?"), we adopt a more nuanced format where models must select which shapes appear in the figure from a list of options. This design increases task complexity and better evaluates shape discrimination skills. To construct these questions, we assign a binary indicator to each shape in the geometric pool, denoting its presence or absence in the figure. We then randomly sample both existing and non-existing shapes to form the question and candidate choices.

**Counting.**   Counting questions focus on determining the number of occurrences of specific shapes within a figure. Using the structured description of the figure, we extract the count of each shape and formulate questions accordingly. To enhance robustness, we also include non-existent shapes and provide zero as a candidate choice.

**Location.**   Location-based questions evaluate spatial perception by requiring it to identify the quadrant (upper-left, upper-right, lower-left, lower-right) containing a target shape, determined by its centroid. Besides absolute positioning, we also incorporate relative positioning queries, e.g., "Where is shape A in relation to shape B?" Candidate choices for these questions are the four quadrants.

**Size.**   These questions evaluate the model's ability to perceive shape sizes, such as horizontal span, vertical span, and area. After randomly selecting a shape, we query one of its size attributes and calculate the ground truth value from the structured description. Incorrect answers are generated by introducing variations around the true value, ensuring that models make precise distinctions. To standardize evaluation, all image sizes are normalized to 1, and numeric intervals between candidate answers are sufficiently large to minimize ambiguity.

**Relationship.**   This aspect explores the geometric relationships between pairs of shapes, such as tangency, parallelism, or inscription. These questions test the model's understanding of spatial interactions and its ability to discern fine-grained geometric details. To construct a QA pair, we randomly sample a relationship from the structured description and ask the model to identify the nature of the interaction between two shapes. To ensure robustness, we include pairs with no relationships and provide "none of the above" as a candidate choice.

**Reference.**   Reference questions reverse the typical query format by providing attributes (e.g., size, location, or count) and asking the model to identify the corresponding shape. This aspect evaluates the integration of multiple attributes for shape identification. For example, we might ask, "Which shape is larger than S?" where S is a randomly chosen anchor shape. The ground truth is selected based on the specified attribute, and three other shapes are included as distractors. This design challenges the model to synthesize information across different attributes and make accurate inferences.

By incorporating different question aspects, we aim to provide a comprehensive evaluation protocol for geometric perception in MLLMs.

### B.4   DETAILS ON QUESTION-ANSWER GENERATION

For each figure, we aim to generate up to three questions per aspect across the six aspects. However, the actual number of questions per aspect is adaptively determined based on the content of the figure. For instance, if a figure contains only one pair of relationship, we can only generate one valid relationship question.

Plausible distractors are carefully designed for each aspect to ensure both relevance and challenge:

1) **Counting/Size**: All four candidate answers are deliberately selected to be numerically proximate yet perceptually distinguishable. For counting, distractors differ from the correct answer by $\pm 1$, while for size estimation, distractors deviate by $\pm 0.15$. This design ensures that distractors remain plausible while still requiring precise discrimination.
2) **Existence**: In existence-related questions, distractors are chosen from shapes that are not present in the figure but are sampled from the shape pool. Conversely, for non-existence questions, distractors include shapes that are actually present, which introduces a realistic source of confusion.

Table 3: Detailed model sources.

| Model | Size | Source |
|---|---|---|
| GPT-4o | - | gpt-4o-2024-11-20 |
| Gemini-2.5-pro | - | gemini-2.5-pro |
| BLIP2 | 3B | Salesforce/blip2-flan-t5-xl |
| InstructBLIP | 3B | Salesforce/instructblip-flan-t5-xl |
| LLaVA-1.5 | 7B | liuhaotian/llava-v1.5-7b |
| LLaVA-1.5 | 13B | liuhaotian/llava-v1.5-13b |
| mPLUG-Owl3 | 7B | mPLUG/mPLUG-Owl3-7B-241101 |
| MiniCPM-V-2.6 | 8B | openbmb/MiniCPM-V-26 |
| GLM-4V | 9B | THUDM/glm-4-9b |
| Mantis-Idefics2 | 8B | TIGER-Lab/Mantis-8B-Idefics2 |
| LLaMA-3.2-Vision | 90B | meta-llama/Llama-3.2-90B-Vision |
| LLaVA-OneVision | 7B | lmms-lab/llava-onevision-qwen2-7b-ov |
| LLaVA-OneVision | 72B | lmms-lab/llava-onevision-qwen2-72b-ov |
| Qwen2-VL | 7B | Qwen/Qwen2-VL-7B-Instruct |
| Qwen2-VL | 72B | Qwen/Qwen2-VL-72B-Instruct |
| Qwen2.5-VL | 7B | Qwen/Qwen2.5-VL-7B-Instruct |
| Qwen2.5-VL | 72B | Qwen/Qwen2.5-VL-72B-Instruct |
| InternVL2.5 | 8B | OpenGVLab/InternVL2_5-8B |
| InternVL2.5 | 78B | OpenGVLab/InternVL2_5-78B |
| InternVL3 | 78B | OpenGVLab/InternVL3-8B |
| InternVL3 | 78B | OpenGVLab/InternVL3-78B |
| G-LLaVA | 7B | renjiepi/G-LLaVA-7B |
| G-LLaVA | 13B | renjiepi/G-LLaVA-13B |
| Math-LLaVA | 13B | Zhiqiang007/Math-LLaVA |
| Math-PUMA | 7B | Math-PUMA/Math-PUMA_Qwen2VL-7B |
| QVQ-Preview | 72B | Qwen/QVQ-72B-Preview |

3) **Location**: A fixed set of four spatial quadrants (upper left, upper right, lower left, lower right) is used across all location questions; only the order of these options varies.

4) **Reference/Relationship**: Distractors are selected from other visible shapes or relationships present in the same figure, thereby maintaining contextual plausibility. When the figure contains an insufficient number of valid alternatives, additional distractors are supplemented with randomly chosen non-existent shapes or relationships, ensuring that each question consistently provides four meaningful options.

## C  DETAILS ON EVALUATION

### C.1  EVALUATED MODELS

This section provides details of the models we evaluated. The models' responses in this paper were all collected in Sep. 2025.

#### C.1.1  CLOSED SOURCE MODELS

**OpenAI GPT.**  We access GPT-4o (OpenAI, 2023) models via OpenAI API. We evaluate gpt-4o-20241120.

**Google Gemini.**  We access Gemini 2.5 Pro (Google, 2023) via Google Cloud. We evaluate gemini-2.5-pro.

Figure 8: User prompt template for evaluation.

### C.1.2 OPEN SOURCE GENERAL-PURPOSE MODELS

We evaluate a variety of open-source models, including BLIP2 (Li et al., 2023b), InstructBLIP (Dai et al., 2023), MiniGPTv2 (Chen et al., 2023a), LLaVA-1.5 (Liu et al., 2024a), LLaVA-OneVision (Li et al., 2024), mPLUG-Owl3 (Ye et al., 2024), InternVL2.5 (Chen et al., 2024c), InternVL3 (Zhu et al., 2025), MiniCPM-V-2.6 (Yao et al., 2024), GLM-4V (GLM et al., 2024), Mantis-Idefics2 (Jiang et al., 2024a), Qwen2-VL (Wang et al., 2024b), Qwen2.5-VL (Bai et al., 2025), LLaMA-3.2-Vision (Meta, 2024). Table 3 shows the names of open-source models available on HuggingFace. Additionally, for MiniGPTv2, we evaluate it directly on the model checkpoints following the instructions in its GitHub repository[3].

### C.1.3 OPEN SOURCE SPECIALIZED MODELS

In addition to evaluating general-purpose open-source models, we also assess several specialized models that have been fine-tuned for mathematical or reasoning tasks. G-LLaVA (Gao et al., 2025) is designed specifically for solving geometric problems, with enhanced capabilities in analyzing spatial and geometric relationships. Math-LLaVA (Shi et al., 2024), on the other hand, focuses on improving textual mathematical problem-solving skills, enabling more accurate interpretation and processing of math-related language. Math-PUMA (Zhuang et al., 2025) builds upon this by refining mathematical reasoning through a structured three-stage training framework. Lastly, QVQ-Preview (Team, 2024a) is an experimental model aimed at advancing visual reasoning, particularly in contexts requiring complex multimodal inference.

### C.2 EVALUATION SCHEME

For evaluation, we format the data into multiple-choice questions and apply conversation templates tailored to each model. The specific user prompt template is illustrated in Figure 8.

Despite explicit instructions for the models to provide only the option letter in their responses, some models deviate by including additional explanations. To address this issue, we developed a robust answer-parsing scheme. Specifically, our method identifies the last occurrence of any standalone option letter in the response string and interprets it as the model's answer. This approach ensures that even if a model includes supplementary information, the final choice is accurately captured.

To validate the reliability of our parsing scheme, we conducted a manual verification process. We randomly sampled 100 responses from each model on GePBench and checked whether the parsing method successfully extracted the intended answers. The results demonstrate an accuracy rate of 99.9%, confirming that our approach is both valid and effective. This high level of accuracy ensures that the evaluation scores are reliable and consistent across all models.

### C.3 DETAILS ON HUMAN EVALUATION

To ensure a robust and reliable human evaluation process, we recruited 25 volunteers who were willing to join in our work. All participants had prior exposure to basic mathematical concepts, equipping them with the foundational knowledge necessary to comprehend and solve geometric problems. This qualification ensured that they were well-suited for the task. The participants were carefully selected to ensure demographic diversity. They were from different ages groups spanning from 20 to 50, and their professions ranged from Mathematics and Computer Science to Sociology and Psychology. Before the task, they were notified that their responses would be recorded and analyzed for academic use.

From GePBench, we randomly sampled 200 questions from both the easy and hard splits and distributed them to the volunteers. To familiarize the participants with the task format, we provided them with several illustrative examples and clear instructions. Notably, the textual instructions given to the volunteers were identical to the prompts used for evaluating MLLMs, as shown in Figure 8. This consistency in instructions allowed for a fair comparison between human performance and model outputs.

Participants were tasked with solving multiple-choice questions based on the provided examples and guidelines. Their responses were collected and analyzed to establish a benchmark for human performance, which served as a critical reference point for assessing the capabilities of MLLMs.

## D ANALYSES ON VISUAL ENCODERS

From Table 2, we can derive the following observations:

**Higher resolution improves detail recognition but impacts spatial accuracy.** Using a higher resolution in the CLIP-ViT encoder enhances geometric perception in most aspects, except for location-based tasks. This is likely due to that more image tokens enables the inclusion of finer image details, benefiting the overall performance. However, it also increases the complexity of positional embeddings, leading to increased difficulty in recognizing spatial positions.

**Different encoders specialize in different aspects.** For instance, CLIP-based models demonstrates superior spatial awareness with high performance on the location task, while SigLIP performs better on existing tasks. Self-supervised encoder DINOv2 is consistently outperformed by language-supervised models across nearly all sub-tasks. These differences may stem from variations in training data and objectives.

**Mixed encoders underperform in geometric tasks.** Contrary to real-world scenarios where combining encoders often yields better results (Tong et al., 2024a; Yang et al., 2024), mixed visual encoders fail to achieve a better result compared to their individual components, especially on questions related to location. This could be due to the larger number of image tokens, which makes the positional embeddings much more complex. Since transformer-based models rely solely on positional embeddings for spatial information, this results in degraded spatial awareness.

## E DETAILS ON LLAVA-GEP

LLaVA-GeP-7B is trained based on the LLaVA-1.5 architecture, which integrates the Vicuna-1.5 language model, CLIP-ViT-L-336 as the visual encoder, and an MLP serving as the mapping layer between the visual and textual modalities. To ensure consistency with the original LLaVA framework, we utilize the pretrain[4] and finetune[5] scripts provided in the official LLaVA codebase.

---

[3] https://github.com/Vision-CAIR/MiniGPT-4

[4] https://github.com/haotian-liu/LLaVA/blob/main/scripts/v1_5/pretrain.sh

[5] https://github.com/haotian-liu/LLaVA/blob/main/scripts/v1_5/finetune_lora.sh

## E.1 Data Preparation and Training Procedures

To train LLaVA-GeP-7B, we construct a large-scale training dataset, GePBench-train, using our specialized data synthesis engine. This dataset comprises over 300K samples, designed to enhance geometric perception capabilities. Following the methodology outlined by the original LLaVA authors, we adopt a two-stage training procedure:

**(1) Feature Alignment Stage**: We extract 150k images from both the easy and hard splits of GePBench-train. For each image, structured textual descriptions are generated using Qwen-2.5-14B (Team, 2024b), which is prompted to produce detailed captions based on the input descriptions. These image-caption pairs are then combined with the original LLaVA pretraining dataset, which includes 558K samples sourced from LAION-CC-SBU. The resulting dataset is used to finetune the MLP mapping layer, ensuring alignment between visual features and textual representations.

**(2) Visual Instruction Tuning Stage**: In this stage, we randomly sample 20k examples from each aspect (e.g., size, location, counting) across both the easy and hard splits of GePBench-train, totaling 240k samples. These samples are reformatted into the LLaVA training schema and merged with the original dataset. The combined dataset is used to jointly finetune the MLP mapping layer and the LLM backbone, enabling the model to better various downstream tasks.

## E.2 Implementation Details

During the pretraining phase, LLaVA-GeP-7B employs a global batch size of 256, a learning rate of 1e-3, and runs for one epoch with a maximum sequence length of 2048. In the fine-tuning phase, the model uses a global batch size of 128, a reduced learning rate of 2e-5, and is trained for one epoch with the same sequence length. Following the optimization strategy of LLaVA, we use the Adam optimizer without weight decay and apply a cosine learning rate schedule with a warmup ratio of 3%. To optimize GPU memory usage, we implement Fully Sharded Data Parallel (FSDP) and gradient checkpointing, avoiding offloading to maximize efficiency. Additionally, BF16 and TF32 precision settings are enabled to strike a balance between computational speed and numerical accuracy.

The training process was conducted on eight A6000 GPUs, each equipped with 48GB of memory. Pretraining completed in approximately 12 hours, while LoRA fine-tuning required 32 hours.

## E.3 Evaluation Details

After training, the LLaVA-GeP-7B model is evaluated on a broad range of general-purpose tasks to comprehensively assess its performance across diverse domains. These evaluations encompass both widely adopted vision-language benchmarks and specialized datasets that target specific downstream capabilities.

**General Benchmarks** Following the standard evaluation protocol used in LLaVA, we evaluate our model on a suite of established vision-question answering (VQA) benchmarks, including VQAv2 (Goyal et al., 2019), GQA (Hudson & Manning, 2019), VizWiz (Gurari et al., 2018), TextVQA (Singh et al., 2019), POPE (Li et al., 2023c), MMBench (Liu et al., 2024b), SeedBench-Image (Li et al., 2023a), MME (Fu et al., 2023), LLaVA-Wild (Liu et al., 2023), and MM-Vet (Yu et al., 2024). These benchmarks are designed to measure the model's general visual understanding and reasoning abilities, as well as its performance across a wide variety of multimodal tasks.

**Specialized Benchmarks** To further probe the model's applicability to domain-specific tasks, we conduct evaluations on expert-curated datasets from various specialized fields:

1. **Scientific Diagrams and Document Interpretation:** This category includes DocVQA (Mathew et al., 2021), CharXiv (Wang et al., 2024d), AI2D (Kembhavi et al., 2016), and ScienceQA (Lu et al., 2022), which test the model's ability to interpret scientific figures, documents, and educational content.

2. **Mathematical Problem Solving:** We use MathVerse (Zhang et al., 2024a), MathVista (Lu et al., 2024b), and We-Math (Qiao et al., 2024) to assess the model's capacity for solving complex mathematical problems involving geometric reasoning and calculation.

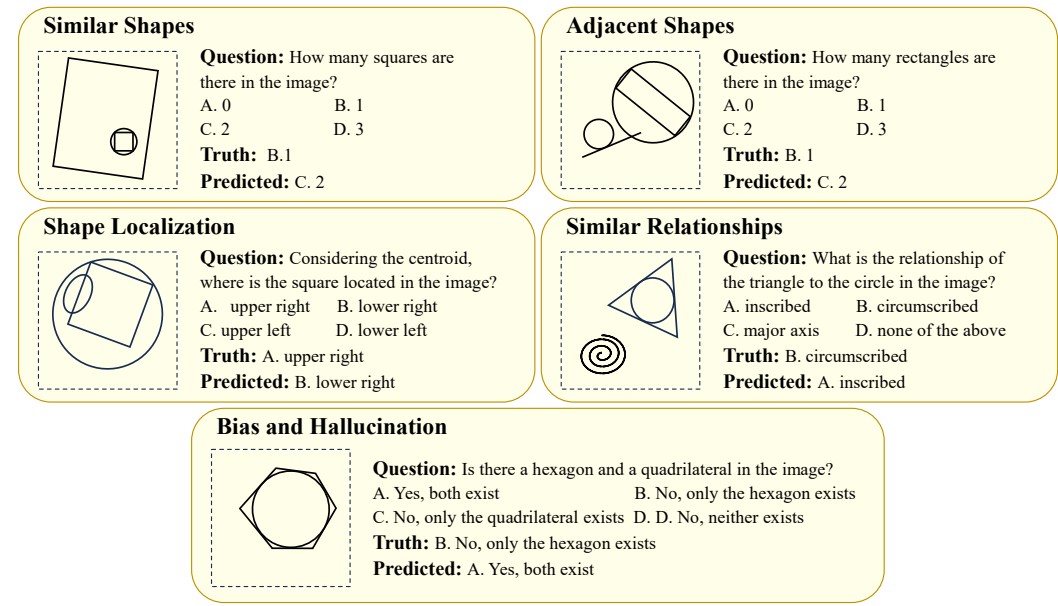

Figure 9: Example questions and model predictions for each error category.

3. **Medical Image Analysis:** To evaluate performance in healthcare-related vision-language tasks, we include PMC-VQA (Zhang et al., 2023), SLAKE (Liu et al., 2021), Path-VQA (He et al., 2020), and MedXpertQA (Zuo et al., 2025), which focus on clinical reasoning and medical imaging interpretation.

For a fair comparison, we train an LLaVA-1.5-7B model from scratch in the same environment and following the same training procedures as LLaVA-GeP-7B, ensuring the only difference between them being the training data. Both models are evaluated on the aforementioned benchmarks using the evaluation scripts from the official codebase. For benchmarks such as LLaVA-Wild, MathVerse, MathVista and CharXiv, we employ Qwen-2.5-14B (Team, 2024b) as the judge model to extract answers from model responses or score them directly according to the guidelines specified for each benchmark. For SLAKE-En and PathVQA, we evaluate the performance on the closed-class data.

## F    DETAILS ON ERROR ANALYSIS

### F.1    PRIMARY ERROR CLASSES

For each of the five primary error classes, we provide an illustrative example in Figure 9, the identified error categories are as follows:

**Failure to discriminate visually similar shapes.**    One of the most prevalent error types observed across nearly all task aspects is the model's inability to distinguish between visually similar geometric shapes. For instance, rectangles are frequently misclassified as squares, and spirals are mistaken for circles. This suggests a limitation in the model's ability to accurately map low-level visual features to high-level abstract geometric concepts. It reflects a potential misalignment between the visual and textual modalities, particularly when fine-grained distinctions are required.

**Misinterpretation of adjacent shapes.**    A widespread issue in the counting and reference aspects is confusion regarding adjacent or spatially overlapping shapes, especially when they are closely positioned. As illustrated in Figure 9, the model identifies two rectangles instead of one, likely due to the proximity of the rectangle's edges to the circle's boundary. This indicates a lack of sensitivity to spatial relationships and fine visual details, which are crucial for accurate geometric perception.

**Inaccurate localization of geometric shapes.** In the context of location-related tasks, a dominant source of error lies in the model's inability to precisely determine the positions of specific shapes within the image. For example, although the centroid of a square lies in the upper-right quadrant of the image, the model might incorrectly place it in the lower-right quadrant. These localization errors highlight deficiencies in the model's spatial perception capabilities.

**Confusion between similar geometric relationships.** For relationship-based queries, many errors arise from the model's difficulty in distinguishing between semantically similar geometric relationships. According to our statistics, the relationships of inscription, circumscription, and tangency are frequently conflated. This mirrors the first error category in that both suggest a weak linkage between visual input and the corresponding geometric abstractions, underscoring a broader challenge in multimodal alignment.

**Bias and hallucination.** Finally, significant issues related to bias and hallucination are observed, particularly in the existence and size aspects. In existence questions, when asked whether two shapes appear independently in the image, the model often generates responses indicating that both shapes either coexist or are entirely absent, regardless of the ground truth. This reflects a strong tendency toward hallucination. Furthermore, in size estimation tasks, we find that 94% of the model's predictions overestimate the true size of shapes, irrespective of their type, pointing to a systematic perceptual bias in how the model interprets geometric scale and proportion.

### F.2 Investigation into Bias and Shape Miscomprehension

To further substantiate out findings on the common failure modes in Section 5.4 with more granular, model-internal evidence, we conduct an in-depth diagnostic study on the most frequent error categroy: Bias and Shape Miscomprehension, employing token probability distribution analysis and saliency maps. Due to the lack of access to internal representations in closed-source models like GPT-4o, this analysis is performed on the open-source models InternVL2.5-8B.

**Analysis on bias via token probability distributions.** To investigate potential biases in model predictions, we analyze the output-layer logits when the model predicts the final option letter (e.g., A, B, C, D). We compute the average predicted probability for each candidate choice pattern across all questions and compare these with the corresponding ground truth frequencies. This allows us to assess whether the model exhibits systematic preference for certain answer patterns independent of input content, indicating learned biases. Each aspect of questions is associated with distinct choice patterns:

1) Size: Questions about length, width, or area, with numerical options from smallest to largest.
2) Counting: Questions asking for the number of shapes, with numerical options from smallest to largest.
3) Existence: Binary co-existence queries (e.g., whether shapes A and B both exist), with three possibilities: both, only one of them, neither.
4) Relationship: Questions about geometric relationships between two shapes, with three valid relations and an "none of the above" option.
5) Location: Questions about the quadrant position of a shape (upper-left, upper-right, lower-left, lower-right).
6) Reference: Questions asking which of four candidate shapes matches a given description.

The comparison of the average probability with ground truth frequency on InternVL2.5-8B is provided in Table 4.

Our results reveal misalignments between the model's predicted probabilities and the actual distribution of correct answers across all aspects. We summarize the bias patterns for each aspect as follows.

1) Size and Counting: the model systematically assigns higher probabilities to larger numerical values even when the real answer is small.
2) Existence: the model shows a preference for positive options, favoring "both exist" against "neither exists" regardless of ground truth.

Table 4: Ground truth frequency and average predicted probability of InternVL2.5-8B (%). Note that predicted probabilities may not sum exactly to 100% due to small probabilities assigned to tokens other than A, B, C or D.

| Aspect | Choice Pattern | Ground Truth Frequency | Predicted Avg. Probability |
|---|---|---|---|
| Size | smallest | 46.4 | 24.6 |
| | smaller | 29.0 | 24.6 |
| | larger | 15.3 | 25.1 |
| | largest | 9.3 | 25.0 |
| Counting | smallest | 36.5 | 32.2 |
| | smaller | 49.8 | 21.8 |
| | larger | 11.3 | 24.3 |
| | largest | 2.5 | 19.0 |
| Existence | both exist | 24.8 | 26.0 |
| | only one exists | 49.6 | 54.9 |
| | neither exists | 25.6 | 19.0 |
| Relationship | valid relationship | 84.2 | 73.2 |
| | none of the above | 15.8 | 22.3 |
| Location | upper-left | 18.9 | 24.7 |
| | upper-right | 25.9 | 23.9 |
| | lower-left | 27.0 | 25.4 |
| | lower-right | 28.1 | 24.6 |
| Reference | Line | 13.2 | 13.5 |
| | Polygon | 33.1 | 52.2 |
| | Circle-like | 53.7 | 33.5 |

3) Relationship, Location and Reference: the model prefers a specific option, i.e., "none of the above" in Relationship, "upper-left" in Location, and polygons in Reference.

These misalignments reveal that current MLLMs do not perceive geometric content in a fully faithful or input-conditional manner. Instead, they rely on heuristic priors, such as numerical magnitude assumptions, positional preferences, or category-level stereotypes, which lead to systematic errors. Such biases undermine model reliability, particularly in domains requiring precise visual understanding.

**Analysis on shape miscomprehension via saliency maps.** To examine cases of miscomprehension, particularly confusion between visually similar shapes, we employ saliency maps derived from attention weights of the last layer in the model during the prediction of the option letter token. This allows us to visualize which regions of the input image the model attends to when making decisions. We analyze 20 correctly answered and 20 incorrectly answered samples from InterVL2.5-8B. Example saliency maps for an incorrect sample and a correct sample are provided in Figure 10. Our findings are as follows:

- In correct predictions, at least 6 attention heads successfully attend to the target shape, indicating focused and accurate visual grounding.
- In incorrect predictions, we observe that in 7 out of 20 cases, the model allocates more attention heads to focus on shapes that are semantically incorrect. For example, the model attends to a line when the correct answer refers to a polygon. These attention patterns suggest that the LLM lacks sufficient discriminative sensitivity to geometric shapes.

These findings align with our error classification in Figure 7, where 38% of all errors are attributed to confusion between similar or adjacent shapes. The saliency analysis provides internal evidence that shape miscomprehension is a significant failure mode.

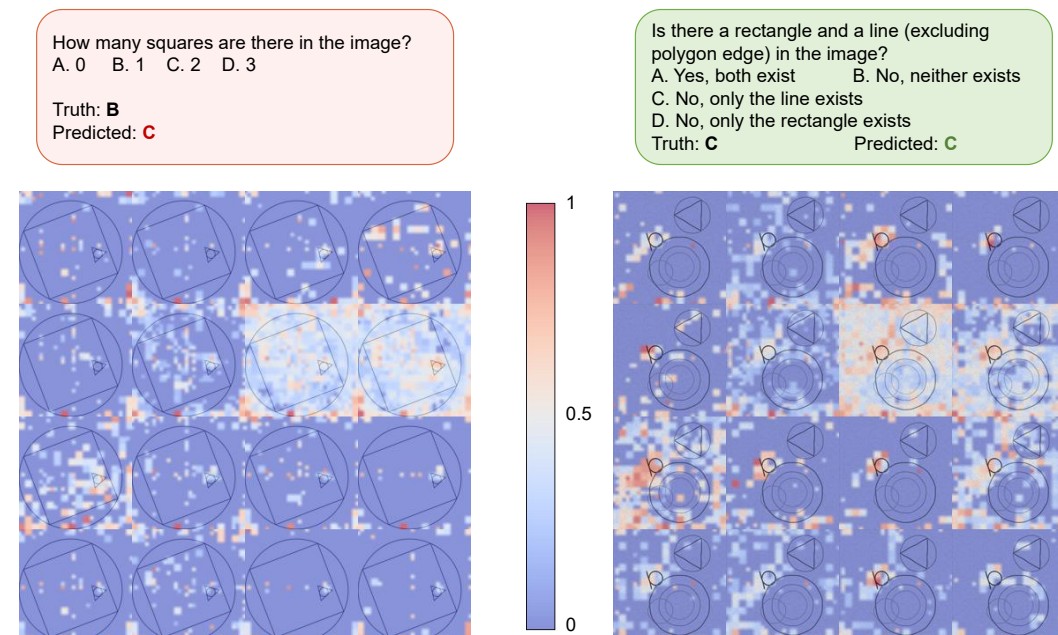

Figure 10: Example saliency maps for an incorrect sample (left) and a correct sample (right). Each tile represents an attention head, and different colors on image patches indicate different values of normalized attention strength. In the incorrect sample, when asked to count squares, the model allocate more attention heads to the triangle and circle instead, resulting in incorrect prediction. In the correct sample, the model allocates most of its attention to the target shape (the line), leading to correct answer.

Table 5: Performance comparison after fine-tuning specific modules.

|  | Average Score |
| --- | --- |
| Original Model | 37.1 |
| + Visual Encoder | 46.5 |
| + Mapping Layer | 46.6 |
| + LLM Backbone | 60.4 |

Table 6: Linear probing results on InternVL2.5-8B across different question types.

| Question Category | Shape | Location | Relation | Size |
| --- | --- | --- | --- | --- |
| Number of Classes | 3 | 4 | 8 | 4 |
| Acc. (Vision Encoder) | 100.0 | 91.8 | 95.3 | 88.8 |
| Acc. (Alignment Module) | 99.8 | 93.5 | 95.3 | 91.8 |
| Acc. (LLM Backbone) | 98.8 | 91.5 | 92.5 | 83.5 |

### F.3 INVESTIGATION INTO SOURCE OF FAILURE

To localize where the failures originate within the model architecture, i.e., visual encoder, vision-language mapping layer, or the LLM backbone, we conduct 2 more experiments: module-specific fine-tuning and probing.

**Module-specific fine-tuning.** To understand whether the error stems from visual encoder, vision-language mapping layer, or LLM backbone, we adopt a module-isolated training strategy on the LLaVA-1.5-7B model: we fine-tune each of the three core components individually on a subset of 10K training samples from GePBench, while keeping the other two components frozen. This ablation allows us to assess the relative potential for improvement in each module, thereby revealing which component contributes most significantly to errors.

As shown in Table 5, fine-tuning only the LLM backbone leads to a substantial performance gain, whereas updating the vision encoder or mapping layer yields only marginal improvements. This indicates that the LLM has the highest headroom for improvement and is likely the dominant source of errors.

**Probing Experiments.** To obtain a more fine-grained understanding of information flow across modules, we conduct probing experiments on the InternVL2.5-8B model. We train linear classifiers

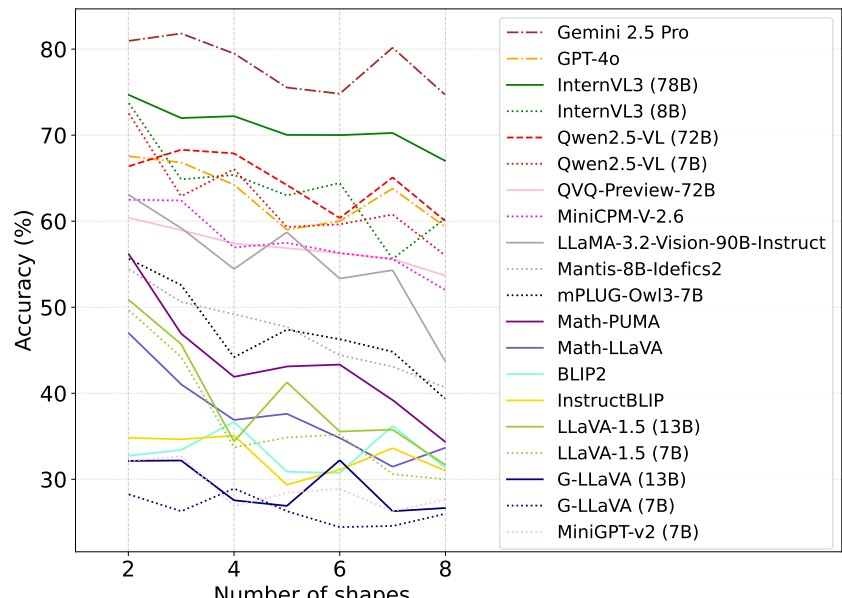

Figure 11: Comparison of the performance of representative models on questions categorized by different number of shapes in the figure.

on the hidden representations produced by each module to evaluate how well geometric information is preserved and accessible at each stage.

We categorize questions into four semantic types, each corresponding to a distinct geometric perception capability:

1) Shape-based: Recognition of basic shape categories (line, polygon, circle-like).
2) Location-based: Identification of spatial position (quadrant classification).
3) Relation-based: Understanding of geometric relationships (e.g., tangency, parallel).
4) Size-based: Perception of shape size (tiny, small, large, huge).

For each category, we construct 1.6K training and 0.4K test samples, each containing synthetically rendered figures paired with labels. A single linear classifier is trained to predict the target class from the module's output representations.

The results in Table 6 show that both the vision encoder and alignment module encode geometric information effectively, with high probe accuracy across all categories. Moreover, the alignment module slightly enhances discriminability in most cases, indicating successful cross-modal integration. However, a noticeable performance drop occurs after the LLM backbone, particularly in size and relation questions. This degradation suggests that the LLM fails to fully utilize or preserve critical geometric information present in earlier stages, implicating the LLM backbone as the major source of errors.

## G  IMPACT OF DATA ATTRIBUTES ON MODEL PERFORMANCE

### G.1  IMPACT OF THE NUMBER OF SHAPES

The number of geometric shapes per figure can significantly affect model performance. A higher number of shapes increases the complexity of the figure and offers more challenges in visual perception. To investigate this, we categorize the questions in GePBench based on the number of shapes and evaluate model performance across these groups.

The results, shown in Figure 11, reveal a general decline in performance as the number of shapes increases. This trend aligns with our expectation that figures with more shapes generally demand a

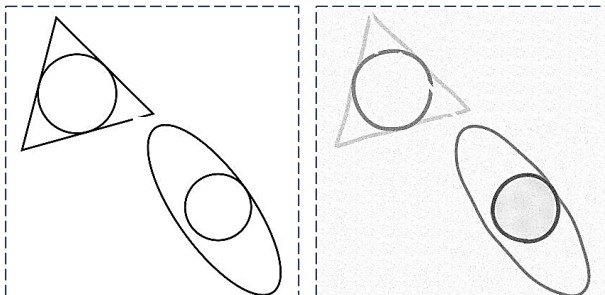

Figure 12: Comparison of figures with and without noise. The left figure does not contain noise and the right one does.

Table 7: Performance difference between noisy samples and noise-free samples of easy split for representative models.

| Model Class | Size | Avg. | Ext. | Cnt. | Siz. | Loc. | Ref. | Rel. |
|---|---|---|---|---|---|---|---|---|
| GPT-4o | - | 66.4 | 77.4 | 70.5 | 16.1 | 61.5 | 88.3 | 84.8 |
| + noise | | 61.6 | 66.2 | 70.0 | 18.1 | 63.4 | 75.2 | 76.4 |
| Gemini-2.5-pro | - | 80.8 | 79.0 | 77.8 | 73.1 | 80.3 | 86.1 | 88.2 |
| + noise | | 79.6 | 71.8 | 73.9 | 80.3 | 80.3 | 86.9 | 84.4 |
| LLaMA-3.2-Vision | 90B | 58.2 | 62.6 | 72.0 | 13.0 | 49.3 | 69.3 | 82.7 |
| + noise | | 58.8 | 61.0 | 68.1 | 14.0 | 54.5 | 72.3 | 83.1 |
| LLaVA-OneVision | 7B | 60.7 | 62.6 | 74.9 | 26.9 | 53.1 | 74.5 | 72.2 |
| + noise | | 61.1 | 61.5 | 74.4 | 30.6 | 50.7 | 73.7 | 75.9 |
| LLaVA-OneVision | 72B | 68.0 | 73.3 | 78.7 | 24.4 | 53.1 | 93.4 | 84.8 |
| + noise | | 67.5 | 72.3 | 77.3 | 24.9 | 56.8 | 92.0 | 81.9 |
| Qwen2.5-VL | 7B | 66.9 | 71.8 | 75.4 | 25.9 | 68.5 | 81.0 | 78.9 |
| + noise | | 67.7 | 71.8 | 74.9 | 28.0 | 72.8 | 82.5 | 76.4 |
| Qwen2.5-VL | 72B | 67.8 | 75.4 | 76.8 | 15.5 | 70.0 | 86.9 | 82.3 |
| + noise | | 68.6 | 75.4 | 74.4 | 20.7 | 75.1 | 86.1 | 79.7 |
| InternVL3 | 8B | 67.7 | 80.0 | 73.4 | 22.8 | 67.1 | 81.8 | 81.0 |
| + noise | | 68.9 | 82.1 | 72.5 | 28.0 | 70.4 | 81.8 | 78.5 |
| InternVL3 | 78B | 73.1 | 75.4 | 76.3 | 50.3 | 65.7 | 86.1 | 84.8 |
| + noise | | 74.4 | 75.9 | 77.8 | 52.3 | 67.1 | 88.3 | 85.2 |

greater capacity for geometric perception. Such findings underscore the need for models to develop stronger foundational visual perception skills to handle more complex geometric inputs.

### G.2 IMPACT OF NOISE

#### G.2.1 NOISE RESILIENCE OF DIFFERENT MODELS

To visualize the add synthetic noise, we provide a side-by-side comparison of figures with identical structures in Figure 12, where the left figure is free from noise and the right one contains noise. We conduct an ablation study to investigate the effect of such noise on model performance. Specifically, noise is introduced to originally noise-free figures of the easy split, and the same questions are applied. Table 7 shows the performance differences for various models after adding noises.

Interestingly, the results reveal that several models, particularly open-sourced ones, achieve improved performance when noise is introduced. The gains are most pronounced in the Size and Location tasks, where nearly all models benefit. We hypothesize that these improvements arise from unintended perceptual cues introduced by noise:

1) Border distortions may cause shape borders to appear expanded, making object boundaries more detectable to visual encoders.
2) Perlin noise inside closed shapes can enhance texture contrast, facilitating size estimation.

Table 8: Average model performance across all six aspects on images of different levels of noise

| Model | Easy | Easy-Noisy | Hand-Drawn | Real-World |
|---|---|---|---|---|
| GPT-4o | 66.4 | 61.6 | 60.0 | 64.6 |
| InternVL2.5-8B | 58.1 | 59.3 | 56.2 | 53.3 |
| InternVL2.5-78B | 71.1 | 72.0 | 69.6 | 69.6 |
| Qwen2-VL-7B | 64.0 | 63.2 | 46.7 | 50.0 |
| Qwen2-VL-72B | 68.5 | 67.6 | 65.0 | 66.7 |
| LLaVA-1.5-7B | 40.6 | 40.8 | 41.2 | 40.0 |
| LLaVA-1.5-GeP | 81.9 | 77.2 | 73.3 | 61.7 |

The variation in other aspects appears to be model-specific. For instance, GPT-4o and Gemini-2.5-Pro exhibit notable performance decline, whereas most others perform comparably to or slightly better than the noise-free condition. This behavior may be attributed to the nature of the training data. While noise introduces additional visual challenges, certain models are more resilient because their training datasets include more scenarios with visual degradation. In some cases, the noisy figures might align more closely with the distribution of the real-world images in training data compared to the original, noise-free figures, particularly when geometric shapes are absent from the training corpus. Notably, the InternVL-3 series show stable improvements across most aspects. We presume that such gains arise from the random JPEG compression augmentation strategy employed during training, which enhances model's robustness to noise.

### G.2.2 EFFECTS OF DIFFERENT LEVELS OF NOISE

To further investigate the effects of noise and extend our analysis to real-world scenarios, we introduced two new data sources with progressively increasing noise level.

1) **Hand-drawn diagrams**: These include 40 sketches manually drawn. They capture natural irregularities in line quality, shape composition, and spatial alignment, reflecting human drafting styles and real-world variations

2) **Real-world scientific diagrams**: We manually selected 40 images with clear geometric content from existing datasets containing real scientific and illustrative figures, including ScienceQA, AI2D, and ChartXiv.

For each source, we constructed 240 high-quality QA pairs following the same procedure of the data engine for GePBench. While similar to the noisy images in GePBench, this subset ensures that the variations are genuinely organic. We then evaluated multiple models (including general MLLMs and our LLaVA-1.5-GeP) on this newly collected dataset, and report their average performance across six aspects.

The findings in Table 8 show a consistent performance drop across all models when transitioning from synthetic to real-world inputs, highlighting the increased challenge posed by naturally occurring visual variations. These experiments underscore substantial limitations in visual perception capabilities of current MLLMs.

Crucially, LLaVA-1.5-GeP significantly outperforms LLaVA-1.5-7B in both synthetic and real-world settings. This suggests that exposure to diverse geometric patterns and spatial understanding tasks during training generalizes to real-world visual scenarios.

## H DETAILS ON THE PERFORMANCE DATA ACROSS VARIOUS BENCHMARKS

To analyze the performance gap between humans and MLLMs, we adopt the difference between GPT-4o and humans as performance gap, as it is widely recognized as one of the most robust and extensively tested models. For benchmarks where GPT-4o results are unavailable, we substitute with GPT-4V performance metrics.

Performance metrics for GPT-4o, GPT-4V, and human benchmarks are primarily sourced from their respective original research papers. For benchmarks without directly reported metrics, we rely on their official leaderboards, as these provide the most up-to-date and reliable results. To maintain consistency, third-party leaderboards are excluded from consideration.

Table 9: Detailed information and data source of different benchmarks.

| Benchmark | Release Date | Employed Model | Source |
|---|---|---|---|
| A-Bench | Jun. 2024 | GPT-4o | Original Paper |
| BLINK | Apr. 2024 | GPT-4o | Original Paper |
| CODIS | Feb. 2024 | GPT-4V | Original Paper |
| HR-Bench | Aug. 2024 | GPT-4o | Original Paper |
| II-Bench | Jun. 2024 | GPT-4o | II-Bench Leaderboard |
| M$^3$CoT | May. 2024 | GPT-4o | M$^3$CoT LeaderBoard |
| MARVEL | Apr. 2024 | GPT-4V | Original Paper |
| MathVerse | Mar. 2024 | GPT-4V | MathVerse Leaderboard |
| MathVista | Oct. 2023 | GPT-4o | MathVista Leaderboard |
| MMMU | Nov. 2023 | GPT-4o | MMMU Leaderboard |
| MuirBench | Jun. 2024 | GPT-4o | Original Paper |
| Q-Bench | Sep. 2024 | GPT-4V | Original Paper |
| UNIAA | Apr. 2024 | GPT-4V | Original Paper |
| VCR | Jun. 2024 | GPT-4o | Original Paper |
| WinoGround | Apr. 2022 | GPT-4V | Winoground Benchmark |

For multilingual datasets or benchmarks with multiple splits that do not provide an overall score, we select the hardest split with the maximum number of samples in English. For instance, we use the HR-Bench-8K split for HR-Bench and the VCR-en-hard split for VCR. Table 9 details the benchmarks, sources, and models employed in our evaluation.

