# OpenReview forum: "GePBench: Evaluating Fundamental Geometric Perception for Multimodal Large Language Models"
_ICLR.cc/2026/Conference — ICLR 2026 Conference Withdrawn Submission_

### Official Review · Reviewer_XVyi · 2025-10-16

**Soundness:** 2
**Presentation:** 2
**Contribution:** 2
**Rating:** 4
**Confidence:** 4

**Summary:**

This paper introduces GePBench, a dataset designed to evaluate multimodal large language models (MLLMs) on tasks involving the understanding of geometric shapes and spatial relationships. The evaluation shows that even state-of-the-art models such as GPT-4o and InternVL3 perform poorly on this dataset. The authors further conduct experiments with LLaVA-1.5-7B trained on GePBench and demonstrate that training on this dataset improves performance on popular benchmarks of vision-language tasks.

**Strengths:**

* This study investigates the geometric perception capability of multimodal large language models (MLLMs), which is a fundamental ability required for effectively solving vision-language tasks.
* As shown in Figure 5, training on the proposed dataset improves the performance of LLaVA-1.5-7B across diverse benchmarks. These results demonstrate that the proposed dataset is useful for developing and improving MLLMs.
* This paper presents an error analysis (Section 5.4) that offers useful insights for future research on improving MLLMs.

**Weaknesses:**

### Framing and Scope

Although GePBench comprises multiple sub-tasks, it primarily focuses on understanding the relative relationships among multiple geometric shapes. This scope is narrower than what the title, “evaluating fundamental geometric perception,” suggests. I recommend revising the descriptions, including the title and abstract, to clarify that the proposed dataset is intended to evaluate relational understanding between geometric shapes rather than general “fundamental geometric perception.” Otherwise, the paper should more clearly define and specify the scope of “geometric perception” within the context of this study.

### Limited Experiments

The finding that MLLMs exhibit limitations in geometric perception (i.e., the main result in Section 4.2) has already been reported in prior work conducted on different but closely related tasks [1, 2]. Therefore, I consider that the improvement achieved through training on the proposed dataset represents the most significant contribution of this paper. However, the experiments supporting this claim are limited in scope.

Specifically, the results are presented only for LLaVA-1.5-7B, an outdated and relatively weak model. As shown in Table 1, stronger models such as InternVL3-8B are available. To provide more convincing evidence for the effectiveness of the proposed dataset, the authors should include evaluations on more recent and capable models. A deeper analysis of these experimental results would also strengthen the paper’s contributions.

### Novelty, Missing Related Work

Prior studies [1, 2] report similar observations that MLLMs perform poorly on geometric perception tasks. Moreover, the limited spatial recognition capabilities of MLLMs have been reported in prior work [e.g., 3, 4]. Given the findings reported in these studies, the results presented in this paper are not surprising, as MLLMs are already known to struggle with geometric and spatial understanding. I am not suggesting that the proposed dataset is entirely without novelty, as it includes certain sub-tasks not covered in existing datasets. However, the paper should discuss why existing datasets are insufficient and how the proposed dataset offers clear advantages. Including such a discussion would help better position the contribution of GePBench within the existing literature.

[1] [Kamoi et al. VisOnlyQA: Large Vision Language Models Still Struggle with Visual Perception of Geometric Information. COLM 2025.](https://openreview.net/forum?id=PYHwlyu2fa)

[2] [Rahmanzadehgervi et al. Vision language models are blind. ACCV 2024.](https://openaccess.thecvf.com/content/ACCV2024/html/Rahmanzadehgervi_Vision_language_models_are_blind_ACCV_2024_paper.html)

[3] [Kamath et al. What’s “up” with vision-language models? Investigating their struggle with spatial reasoning. EMNLP 2023.](https://aclanthology.org/2023.emnlp-main.568/)

[4] [Cheng el al. SpatialRGPT: Grounded Spatial Reasoning in Vision-Language Models. NeurIPS 2024.](https://neurips.cc/virtual/2024/poster/95720)

**Questions:**

I would appreciate responses from the authors to the points mentioned in the Weaknesses section.

---

### Official Review · Reviewer_iPrg · 2025-10-20

**Soundness:** 3
**Presentation:** 3
**Contribution:** 2
**Rating:** 4
**Confidence:** 3

**Summary:**

This paper presents a focused study on geometric perception (i.e. perceiving and understanding precise details of geometric shapes) of MLLMs. To test these capabilities, the authors propose GePBench, a dataset procedurally generated from a set of predefined rules, and show that frontier open- and closed-source MLLMs struggle to perform meaningfully on low-level geometric perception. The authors further show that improvements derived from training on their dataset can transfer to real-world applications and other general-purpose benchmarks.

**Strengths:**

- A key strength of this work is GePBench. With 285K samples, it is a massive and valuable new resource for diagnosing a well-known but poorly quantified weakness in MLLMs. The authors' schematic categorization into six perceptual aspects is a well-considered approach.
- The authors conducted extensive analysis into 27 open- and closed-source MLLMs to demonstrate the prevalence of the struggle to perceive low-level geometric shapes. The detailed analyses are also appreciated.
- Interestingly, the authors show that training on synthetic, geometric perception tasks can in fact improve on domains that require low-level perception in general, as validated by general-purpose benchmarks. This is a nice case of transfer learning.

**Weaknesses:**

- I believe the datasets and analyses are well done. My primary concern stems from missing references to highly relevant prior works. For example [1] (posted to arxiv in December 2024) also identifies that frontier MLLMs struggle in low-level geometric perception tasks and propose a series of training strategies to improve geometric perception performances. Given the high overlap between the two works, the authors should acknowledge and discuss their contributions more clearly.

[1] https://arxiv.org/abs/2412.08737

**Questions:**

I'm particularly interested in the transfer learning experiments. Could the authors provide some qualitative examples that Llava-GeP yield the correct answer whereas the base model does not, and assess whether these improvements are indeed derived from better geometric perception?

---

### Official Review · Reviewer_CUxV · 2025-11-01

**Soundness:** 3
**Presentation:** 3
**Contribution:** 2
**Rating:** 4
**Confidence:** 4

**Summary:**

This paper presents GePBench, a large-scale benchmark specifically designed to evaluate the geometric perception abilities of MLLMs, including their capacity to recognize shapes, spatial relationships, and configurations. GePBench is composed of 80K figures and 285K questions across six aspects (existence, counting, size, location, reference, and relationship).

Extensive evaluations on 27 leading MLLMs reveal that state-of-the-art MLLMs, especially open-sourced ones, face considerable challenges in GePbench. In addition, size and locations are generally more challenging than other aspects of geometric visual perceptions.

The authors further study the impact of different visual encoders, and find that higher resolution enhances fine-grained recognition but sacrifices spatial accuracy, different visual encoders excel in different aspects and mixing visual encoders are not generally beneficial for the geometric perception tasks.

Lastly, the authors train LLaVA-GeP, an enhanced model fine-tuned on GePBench training data, which achieves notable gains on downstream tasks including medical imaging and chart understanding—demonstrating that stronger geometric perception directly benefits broader multimodal understanding and reasoning.

**Strengths:**

1. The explicit isolation of geometric visual perception makes the scope, discussion and conclusion generally clear and focused in this paper, offering concrete insights for advancing multimodal perception research.
2. By combining the GepBench-training dataset, the proposed LLaVA-GeP-7B model achieves consistent improvements over the original LLaVA-1.5-7B across diverse downstream tasks, empirically validating that enhanced geometric perception benefits general multimodal understanding.

**Weaknesses:**

1. The analysis in Section 5.1 reveals several encoder-level inconsistencies that remain underexplored. For instance, the finding that CLIP-ViT-L-336 improves fine-grained recognition but compromises spatial accuracy appears conceptually contradictory, since both rely on precise local encoding. A more detailed examination of positional embedding design/resolution-induced token interactions would be helpful. Similarly, the findings that different visual encoders exhibit complementary strengths and the underperformance of mixed-encoder configurations seem also contradictory to each other. A more detailed examination of positional embedding design/resolution-induced token interactions would be helpful.
2. Some experimental results are counter-intuitive and need additional error diagnosis. Notably, GPT-4o underperforms BLIP-2 on size-related questions despite having substantially stronger overall visual understanding. Such a result makes the validity of the current dataset questionable, a deeper error analysis is needed for understanding this phenomenon and justifying the reliability of the dataset.
3. Although LLaVA-GeP-7B demonstrates meaningful improvements across downstream benchmarks. It is based on an early-stage MLLM (LLaVA-1.5-7B) comparing with recent high-performing models such as Qwen2.5-VL or InternVL-3. Evaluating the same geometric-perception training on these newer model families would provide stronger validation of its universality and clarify whether the observed benefits reflect fundamental inductive bias improvements or model-specific compensations.
4. Additional discussion of related works is needed. A recent work Euclid [1] has a very similar scope (evaluation geometric visual perception of MLLMs) but did a more comprehensive study on more MLLM components. It is encouraged to discuss the distinctions and relations between the findings of these two papers.

[1] Zhang, J., Liu, O., Yu, T., Hu, J., & Neiswanger, W. (2024). Euclid: Supercharging multimodal llms with synthetic high-fidelity visual descriptions. arXiv preprint arXiv:2412.08737.

**Questions:**

1. I’m wondering if the authors have any insights about why the LLaVA-GeP model improves on natural image understanding tasks like GQA, LLaVA_W when augmented only on geometry figures.
2. It is also important to clarify the contradicted/counter-intuitive findings pointed in weakness 1 and 2.

---

### Official Review · Reviewer_gFLM · 2025-11-01

**Soundness:** 2
**Presentation:** 3
**Contribution:** 2
**Rating:** 2
**Confidence:** 4

**Summary:**

The paper introduces a synthetic benchmark for geometric perception—recognizing shapes and their spatial relations—built via a rule-based data engine that turns structured descriptions into rendered figures and MCQs across six aspects (location, size, existence, counting, reference, relationships). The dataset reports 80K images and 285K questions with easy/hard splits, and a finetuned model (LLaVA-GeP) trained on the same engine’s data. Overall, the paper is quite easy to follow and have fair contributions regarding creating a synthetic dataset, and more interestingly has slightly improved performance when finetuned with that. . However, the idea of synthetic data generation itself is not novel, as ver similar methodologies have been explored in prior works in this area, which the paper unfortunately does not cite.

**Strengths:**

1. Clear task focus on low-level geometric perception and a large synthetic set with systematic coverage of six axes.
2. More interestingly, the synthetic generated data seems to slightly improve performance.
3. Fairly comprehensive analysis, showing which aspects of geometric perception is most challenging. That can be helpful for future works in this area.

**Weaknesses:**

1. My primary concern stems from the fact that, the core recipe (shape/attribute pool → rendering →questions) closely resembles prior VQA works; closely related recent work (e.g., VisOnlyQA on geometric perception) and classic precedents (e.g., CLEVR). None of these works seem to be acknowledged in the paper. The paper needs to consider these works and establish the differences of the synthetic generation method to them.

2. Model choice for finetuning: the positive transfer is interesting, but results hinge on an older baseline (LLaVA-1.5-7B). To substantiate generality, newer VLMs (e.g., Qwen3-VL/InternVL-3.5 class) should be finetuned and reported. I am not sure whether these findings will translate to newer VLMs.

3. While the storytelling of this paper revolves around the synthetic data, because these techniques have been used before, I'd suggest rewriting the story to focus on the model improvement side more if the authors notice that improving performance trend in newer models as well.

**Questions:**

1. Figure 2’s description is too small to read; enlarge/clarify the pipeline diagram. Can you enlarge it?
2. Have you covered the following works? I think that might change the focus of the paper:

VisOnlyQA: Kamoi, R., Zhang, Y., Das, S.S.S., Zhang, R.H. and Zhang, R., 2024. Visonlyqa: Large vision language models still struggle with visual perception of geometric information. arXiv preprint arXiv:2412.00947.

Johnson, J., Hariharan, B., Van Der Maaten, L., Fei-Fei, L., Lawrence Zitnick, C. and Girshick, R., 2017. Clevr: A diagnostic dataset for compositional language and elementary visual reasoning. In Proceedings of the IEEE conference on computer vision and pattern recognition (pp. 2901-2910).

---

### Note · Authors · 2026-01-05

I have read and agree with the venue's withdrawal policy on behalf of myself and my co-authors.